# Preventive Therapy for Contacts of Drug-Resistant Tuberculosis

**DOI:** 10.3390/pathogens11101189

**Published:** 2022-10-15

**Authors:** Yousra Kherabi, Simone Tunesi, Alexander Kay, Lorenzo Guglielmetti

**Affiliations:** 1Centre d’Immunologie et des Maladies Infectieuses, Sorbonne Université, INSERM, U1135, Cimi-Paris, Équipe 2, 75013 Paris, France; 2Department of Infectious Diseases, Hôpital Bichat-Claude Bernard, Assistance Publique Hôpitaux de Paris, 75018 Paris, France; 3Department of Infectious Diseases, AON SS Antonio e Biagio e Cesare Arrigo, 15121 Alessandria, Italy; 4Global Tuberculosis Program, Baylor College of Medicine, Houston, TX 77030, USA; 5Centre National de Référence des Mycobactéries et de la Résistance des Mycobactéries aux Antituberculeux, Laboratoire de Bactériologie-Hygiène, Hôpital Pitié-Salpêtrière, APHP Sorbonne Université, 75013 Paris, France

**Keywords:** tuberculosis, multidrug-resistant, treatment, prevention, contacts

## Abstract

Preventing the progression of a drug-resistant tuberculosis (DR-TB) infection to disease is an important pillar of the DR-TB elimination strategy. International guidelines have recently proposed fluoroquinolones for tuberculosis preventive therapy (TPT) in DR-TB contacts, although the available evidence is low quality. The pooled data from small observational studies suggest that a fluoroquinolone-based TPT is safe, effective and cost-effective as a preventive treatment in DR-TB contacts. Three clinical trials are currently ongoing to generate higher quality evidence on the efficacy of levofloxacin and delamanid as a DR-TB preventive therapy. Additional evidence is also needed, regarding TPT treatment in fluoroquinolone-resistant-TB contacts, patient and health care worker perceptions on DR-TB preventive therapy for contacts, and the service delivery models to increase DR-TPT access. This state-of-the-art review presents the current literature on TPT for contacts of DR-TB cases, focusing on the available evidence and international guidelines.

## 1. Introduction

Rifampin-resistant tuberculosis (RR-TB) and multidrug-resistant tuberculosis (MDR-TB), defined as TB resistant to both rifampin and isoniazid, are a global public health issue [1]. In 2019, there were 465,000 incident cases of RR/MDR-TB. The estimated global treatment success rate for RR/MDR-TB is low, around 59% in 2018 [2]. Hence, preventive actions, such as developing effective vaccines and implementing tuberculosis preventive therapy (TPT), are essential to reduce the cases of drug-resistant TB (DR-TB) and achieve the goals of the End TB Strategy, set by the World Health Organization (WHO) [3].

A TB infection occurs when *Mycobacterium tuberculosis* is inhaled without causing clinical signs. Once this happens, a TB infection can be cleared by the innate immune system and remain silent—what we define as a latent tuberculosis infection (LTBI)—or evolve into active TB disease [4]. Active TB is typically associated with the development of clinical and/or radiological changes. A person with active pulmonary TB disease can transmit *M. tuberculosis*. It has been recently estimated that about a quarter of the global population is infected with TB; the rate of infected populations varies considerably across the WHO regions [5]. The reservoir of potential TB cases must be managed in order to control the TB pandemic; this is also true for DR-TB cases, which are linked most frequently with the direct transmission of DR-TB strains, rather than with an acquired drug resistance [6,7,8].

Preventing the progression of a DR-TB infection to disease is thus a major component of the strategy leading to TB elimination [9,10]. Timely screening and treatment of close contacts of patients with DR-TB is a major priority, as 90% of active cases among MDR-TB contacts occurs within the first two years after the exposure [11,12]. This is particularly true for young children and people living with HIV, who have a higher risk of infection with *M. tuberculosis* and a rapid progression towards TB disease [13,14]. Despite the need for DR-TB prevention strategies, global TPT coverage remains insufficient. Between 2018 and 2020, only 29% of children under 5 years and 1.6% of other household contacts—of drug-susceptible and DR-TB cases—eligible for TPT, received it [2]. In addition, the proportion of DR-TB household contacts receiving TPT is presumed to be substantially lower than this overall proportion.

Moreover, the COVID-19 pandemic led to a concerning decrease in DR-TB diagnosis, treatment and contact tracing, resulting in a potential increase in the transmission [2]. The consequences of the pandemic on DR-TB epidemiology are still not fully known, but they underscore the need for a stronger global policy on DR-TB preventive therapy.

The objective of this article is to provide an up-to-date review of the literature on the preventive therapy for contacts of DR-TB cases, focusing on the available evidence, ongoing clinical trials and existing guidelines.

## 2. Published Evidence on the Preventive Therapy for DR-TB Contacts

### Contacts of DR-TB Cases

Household contacts of a patient with active pulmonary TB are at high risk for a TB infection and disease as they have a prolonged exposure to the index cases [15,16]. A meta-analysis published by Shah and colleagues, showed that 47% (95% confidence interval (95% CI), 30–61%) of the DR-TB patients’ household contacts are infected [12]. The prevalence of a TB infection and disease is particularly high among children who are household contacts and exposed to RR-TB, reaching up to 57% in an observational study published by Kim and colleagues [17]. These transmission rates are even higher between mother and children in the household.

Household contact evaluations entail actively detecting and treating contacts with LTBI, who would benefit from TPT. Although considerable that differences exist between countries, this strategy is generally well-established in low-incidence TB settings. Conversely, in high-burden, low-resources settings, a less expensive “passive” contact management strategy is the rule [18]. The programmatic surveillance of the DR-TB patients’ contacts without TPT administration, is useful to detect active TB disease at an early stage but it does not curb the risk of developing active TB disease [19]. In a modelling study published by Dodd and colleagues, the active household contact management in rifampicin-susceptible tuberculosis, required more effort to prevent each death, compared to the RR/MDR-TB contact management [20]. This finding suggests that active DR-TB household contact management should be considered a high priority in the global fight against tuberculosis.

## 3. Preventive Therapy: Effectiveness, Safety and Cost-Effectiveness

Multiple TPT regimens are available for drug-susceptible TB. The World Health Organization’s recommended regimens rely on single drugs (6–9 months of daily isoniazid or 4 months of daily rifampin), or multiple medications (3 months of daily isoniazid and rifampin, 3 months of weekly isoniazid and rifapentine and 1 month of daily isoniazid and rifapentine). These regimens, according to the TB pharmacologic principles, are unlikely to provide effective TPT for a MDR-TB infection [5]. However, a prospective study led in Peru by Huang and colleagues has shown some efficacy of TPT with isoniazid, even in contacts of MDR-TB cases [21]. Isoniazid monotherapy is also used as a control in a randomized, controlled trial for TPT of MDR-TB contacts (PHOENIx trial, NCT03568383). While other evidence suggests that isoniazid provides insufficient protection following an infection with MDR-TB, these recent data suggest that isoniazid may need to be re-evaluated in this context.

Several regimens have been tested as DR-TB preventive therapy, using one, two or three drugs thought to be effective against the source case, including isoniazid, pyrazinamide, ethambutol, fluoroquinolones and/or ethionamide. The strong evidence-based policies for TPT of DR-TB contacts are lacking because the current published data primarily consists of small cohort studies. No randomized, controlled trial data is available to date. The results of the main studies analyzing the effectiveness of DR-TB preventive therapy, which were published at the time of writing (September 2022), are described in Table 1. Although the highest proportion of studies are from South Africa, in which the co-prevalence of DR-TB and HIV is high, the largest studies are from Pakistan and Peru. All studies reported here include a pediatric population. The fluroquinolone-based regimens, either a fluoroquinolone monotherapy or fluoroquinolone with a companion drug, e.g., ethambutol or ethionamide, are the most commonly evaluated. The most frequently reported primary outcome is an incidence of active TB disease, and the completion rates are reported in most of studies. Unfortunately, the safety data is not always explicitly detailed.

In 2017, Marks and colleagues published a meta-analysis pooling of the data of the available observational studies on DR-TB preventive therapy [22]. They estimated a statistically significant reduction in TB incidences among the treated, compared to the untreated contacts. In a negative binomial model, they found a 90% (very wide CI, 9–99%) risk reduction of developing active TB. In this meta-analysis, the most effective regimen seemed to include fluoroquinolone combined with ethionamide. Recently, Malik and colleagues published a large cohort study evaluating TPT with a fluoroquinolone-based, 2-drug regimen, in Pakistan [23]. High-risk household contacts of all ages began a 6-month standardized course of preventive therapy with levofloxacin or moxifloxacin, associated with either ethambutol or ethionamide, regardless of the drug susceptibility pattern of the strain harboured by the source case. The control group was an historical cohort of untreated contacts from previous published studies. Overall, 172 adults and children contacts of fluoroquinolone-susceptible DR-TB patients were included. In this cohort, the fluoroquinolone-based therapy reduced the risk for TB disease in high-risk contacts exposed to DR-TB by 65% within 2 years after the diagnosis of a DR-TB in the index patient.

An important factor to consider while evaluating TPT effectiveness is the treatment completion. The extended duration of the therapy, the excessive pill burden and medication related adverse events can contribute to the patient’s decision making surrounding the TPT initiation and completion. In the aforementioned meta-analysis, the overall treatment completion rate was of 68% (95% CI, 64–71%), of all treatment regimens taken into account.

A major reason for the treatment discontinuation, is the toxicity associated with TPT. In a small prospective cohort study performed in the United States of America, Adler-Shohet and colleagues followed, prospectively, 31 children with a suspected MDR-LTBI, for 2 years [28]. Twenty-six children received a 9-month regimen of levofloxacin and pyrazinamide. All of the treated children experienced adverse events, and 42% of them stopped treatment. The severity of the adverse events was not specified in the article, but the most frequent were arthralgia, myalgia, abdominal pain, and elevated hepatic enzymes. Another study evaluating levofloxacin and pyrazinamide as TPTs for a presumptive MDR-LTBI in adults reported similar results, namely that all participants experienced at least one adverse effect, and all discontinued the treatment [34]. In the meta-analysis by Marks et al., regimens containing pyrazinamide seemed to have the highest rate of adverse events (average, 66%) and most of those resulted in the discontinuation of the TPT (average, 51%; 95% CI, 44–59%) [22]. Fluoroquinolones and pyrazinamide should not be used together for TPT. While fluoroquinolone–based regimens (without pyrazinamide) have shown a high percentage (33%) of adverse events, they rarely lead to TPT discontinuation (2%; 95% CI, 1–4%), suggesting a lower severity [22]. While some concerns regarding the safety of fluoroquinolones remain, particularly in the pediatric population, there is now extensive evidence for their safety, including in children [35].

Moreover, the fear of enhancing the emergence of drug resistance has been cited by physicians as an obstacle to TPT prescription [36]. Though this concern has been repeatedly proven to be unfounded for isoniazid regimens in drug-susceptible TB prevention, some authors argue that the absence of an observed elevation in the risk of DR-TB among those receiving TPT, in small studies, does not imply that the selective pressure imposed by a community-wide TPT will not be substantial, especially in the context of DR-TB contacts [37,38].

Overall, the existing data suggest that TPT for a DR-TB exposure and infection is both effective and safe. In a modelling study, Dodd and colleagues assessed the cost-effectiveness of screening child household contacts of RR/MDR-TB patients and compared the TPT regimens given to the groups who were at high risk of developing the tuberculosis disease [20]. The results indicated that TPT given to children younger than 15 years is likely to be cost-effective for preventing new TB cases and reducing TB-associated mortality in most countries. Moreover, a regimen with levofloxacin was more cost-effective than a delamanid-based TPT. Other modelling studies came to similar conclusions regarding the TPT of a MDR-LTBI: Holland and colleagues found that the association of fluoroquinolone with ethambutol was cost-effective, while Fox and colleagues’ model found that fluoroquinolone monotherapy was cost-saving [39,40]. In the aforementioned meta-analysis, the MDR-LTBI preventive therapy was cost-saving, compared to no treatment [22]. The most cost-effective regimen was fluoroquinolone associated with ethambutol, followed by fluoroquinolone alone, and the association of pyrazinamide with ethambutol. In the specific population of young contacts, in patients with comorbidities or in high tuberculosis burden settings, a monotherapy with fluoroquinolone became more cost-effective.

## 4. Available Guidelines on Preventive Therapy for DR-TB Contacts

The recommendations from available international guidelines are summarized in Table 2. Although based on the low-level evidence coming from the observational studies, the guidelines generally agree on the need to treat the contacts of MDR-TB patients. A daily levofloxacin-based treatment is the most commonly recommended regimen for both adults and children. The WHO guidelines recommend a fluoroquinolone-based regimen (with or without a companion drug e.g., ethionamide and/or ethambutol), when tolerated. However, the role of companion drugs in TPT regimens is not clear yet, as their use is linked with the increased risk of side effects and early treatment interruption [5]. The European Centre for Disease Prevention and Control (ECDC) guidelines, while still recommending TPT as a first option, underline the lack of strong evidence both in favour and against the pharmacological treatment and allow the strict observation for the onset of the symptoms among contacts [41]. Doctors Without Borders (MSF) guidelines recommend assessing the indication for treatment according to each case, evaluating the resistance patterns of the source case, the risk of progression, the intensity of exposure and the risk of side effects during treatment [42]. If eligibility is established, a six-months standard levofloxacin treatment is recommended. Similarly, the joint American Thoracic Society, the U.S. Centres for Disease Control and Prevention, the European Respiratory Society, and the Infectious Diseases Society of America (ATS/CDC/ERS/IDSA) guidelines recommend to use a daily levofloxacin monotherapy for 6 to 12 months [43]. The use of other drugs, such as pyrazinamide, is not advised due to the frequent discontinuation for toxicity. Overall, all guidelines underline the need for research on TPT in drug-resistant TB contacts, considering the scarcity of the randomized trials and the weakness of the existing evidence in favour of or against TPT in this field.

## 5. Trial Landscape

In recent years, the need to increase the evidence on the efficacy of DR-TB preventive therapy has been satisfied by the launch of three randomized prospective trials, whose primary endpoints are the description of incidence of active TB among patients receiving experimental treatment vs. patients receiving a placebo or standard-of-care treatment (Table 3).

The V-QUIN MDR-TB trial is a placebo-controlled, double-blind trial aiming to assess the role of levofloxacin in TPT for adult household contacts of MDR-TB patients (ACTRN12616000215426) [44]. The population of the trial, performed in Vietnam, included patients aged >15 years who were in contact with confirmed MDR-TB cases, during the prior three months and had a positive tuberculin skin test (TST). The patients were randomized into two groups, one receiving a daily dose of 750 mg of levofloxacin, the other receiving a placebo. The study follow-up was up to 30 months after the start of the treatment. The total expected sample size was 3344 participants. The recruitment for the V-QUIN MDR-TB trial has been recently completed.

The TB-CHAMP (Tuberculosis child multidrug-resistant preventive therapy, ISRCTN92634082) Phase III, cluster randomised controlled trial differs from the V-QUIN MDR TB trial, primarily for the inclusion criteria [45]. This study, set in South Africa, recruits only children aged <5 years who have been in contact with confirmed MDR-TB cases. The focus on children was justified by the lack of evidence in this population, the need to assess the pediatric safety of a fluoroquinolone-based treatment and the increased risk of progression to the active disease in children. The children were randomized into two treatment groups, receiving either 15–20 mg/kg/day of levofloxacin or a placebo, for 24 weeks. The participants were followed up for 18 months after treatment. Aiming to include 1556 patients, recruitment started in 2017 and it is still ongoing.

PHOENIx (Protecting Households On Exposure to Newly diagnosed Index Multidrug-Resistant Tuberculosis Patients) is an ACTG/IMPAACT collaborative phase III, open-label, multicentre, international trial (NCT03568383). The aim is to evaluate the efficacy and safety of delamanid for TPT in high-risk household contacts of MDR-TB. The patients from 12 countries (Botswana, Brazil, Haiti, India, Kenya, Peru, Philippines, South Africa, Tanzania, Thailand, Uganda, Zimbabwe) are separated into two groups, one receiving 26 weeks of delamanid, while the others receive 26 weeks of isoniazid. The patients are followed up for 96 weeks post-randomization. The total recruitment is still ongoing, aiming for a total sample size of 5610 participants.

The results of these three clinical trials are pending. Further research should also focus on individualised treatment strategies, based on the strain of the index case, particularly in the settings with high background rates of a fluoroquinolone resistance [46]. Moreover, recent studies on mouse models have demonstrated the promising potential of short-term regimens, based on newly approved drugs, such as bedaquiline, which should be evaluated as a possible candidate in DR-TB preventive therapy [47].

## 6. Conclusions

High-quality data on the management of TPT in contacts of DR-TB patients are lacking. Three major randomized controlled clinical trials are undergoing and may shed light on the optimal approach to a DR-TB prevention. While waiting for the results of these promising clinical trials, the pooled data of the observational studies show that fluoroquinolone-based TPT regimens are safe, effective and cost-effective in DR-TB contacts. Indeed, all major guidelines recommend TPT for contacts of RR/MDR-TB patients, mostly with fluoroquinolone or with the combination of fluoroquinolone and another drug. Yet, major challenges are still to be faced. First, there are no approved recommendations for TPT for patients in contact with pre-XDR or XDR cases. There is thus an urgent need to evaluate the TPT regimens that do not include fluoroquinolones, in particular those based on new drugs, such as delamanid—with the ongoing PHOENIx trial- and bedaquiline. Second, the TPT treatment completion is also a major concern. All ongoing clinical trials are testing 6-month regimens, which may be perceived as too long for symptom-free contacts and entail a high pill burden, especially for patients who are already treated for another medical condition, such as HIV. Studies on shorter regimens are needed, building on the recent breakthroughs in TPT for a drug-susceptible LTBI. Third, safety concerns can also be a barrier to TPT implementation. Framing a pharmacovigilance network, offering a standardized follow-up for contacts starting TPT and providing counselling for both contacts and physicians, may increase the global uptake of TPT. Finally, individualised treatment strategies, based on the strain of the index case, should be considered for the exposed household contacts of DR-TB patients.

## Figures and Tables

**Table 1 pathogens-11-01189-t001:** Characteristics of the main studies assessing the tuberculosis preventive therapy effectiveness in drug-resistant tuberculosis contacts.

Study	Country	Study Design	DR-TB Contacts Included	Adults/ Children	Evidence of a TB Infection (LTBI)	Compared TPT (Months/Drug)	Primary Endpoint	Grade 3 or 4 Adverse Events	Completion Rate
Gureva et al., 2022 [24]	Russia	Prospective cohort study	72	Children ≤ 18 years-old	LTBI was diagnosed in 51 children, some children were treated without any evidence of a LTBI.	9 Mfx OR 9 Ofx OR No treatment	Incidence of TB disease: 0/58 (0%) with TPT 1/14 (7%) without TPT	None	90%
Malik et al., 2020–2021 [23,25,26]	Pakistan	Prospective cohort study	800	Adults & children	LTBI was diagnosed in six subjects, some subjects were treated without any evidence of a LTBI.	6 Lfx + E OR 6 LFx + Eth OR 6 Mfx + E OR 6 MFx+ Eth	Overall effectiveness on TB incidence compared to the historical control cohorts: 65% (95% CI 13–86)	None	70%
Huang et al., 2020 [27]	Peru	Prospective cohort study	652	Children ≤ 19 years-old	LTBI status was assessed in all subjects, the proportion of subjects included with a proven LTBI is not reported.	6–9 H No specific DR-TB contact control group	Incidence of TB disease 26/652 (4%) with TPT	NR	NR
Adler-Shohet et al., 2014 [28]	United States of America	Retrospec-tive cohort study	31	Children	All of the children included in the study had a proven LTBI.	Lfx + Z	Incidence of TB disease: 0/26 (0%) with TPT 0/5 (0%) without TPT	NR	58%
Bamrah et al., 2014 [29]	Federated States of Micronesia	Prospective cohort study	119	Adults & children	All of the subjects included in the study had a proven LTBI.	12 Lfx 12 Lfx + E 12 LFx + Eth 12 Mfx 12 MFx + E No treatment	Incidence of TB disease: 0/104 (0%) with TPT 3/15 (20%) without TPT	None	83–100%
Garcia-Prats et al., 2014 [30]	South Africa	Retrospec-tive cohort study	24	Children ≤ 15 years-old	LTBI was diagnosed in eight subjects, some subjects were treated without any evidence of a LTBI.	6 H + E + Ofx	Incidence of TB disease: 0/24 (0%) with TPT	None	88%
Seddon et al., 2013 [31]	South Africa	Prospective cohort study	186	Children ≤ 5 years-old HIV-positive children ≤ 15 years-old	LTBI was diagnosed in 73 children, some children were treated without any evidence of a LTBI.	6 HE + Ofx	Incidence of TB disease: 6/186 (3%) with TPT	7/186 (4%)	76%
Denholm et al., 2012 [32]	Australia	Retrospec-tive cohort study	49	Adults & children	All of the subjects included in the analysis had a proven LTBI.	6–9 Mfx +/− E 6 Cfx +/− Z 6 RZE 9 HZ 6–9 RZ No treatment	Incidence of TB disease: 0/11 (0%) with TPT 2/38 (5%) without TPT	None	82%
Schaaf et al., 2002 [33]	South Africa	Prospective cohort study	105	Children ≤ 5 years-old	LTBI was diagnosed in 70 children, some children were treated without any evidence of a LTBI.	6 HZ + Eth 6 HZE 6 HE + Eth 6 E + Eth 6 HZE + Eth 6 ZE + Eth 6 HZ + Eth	Incidence of TB disease: 2/41 (5%) with TPT 13/64 (20%) without TPT	NR	NR

Abbreviations. DR-TB, drug-resistant tuberculosis (i.e., rifampicin-resistant or multidrug-resistant tuberculosis); LTBI, latent tuberculosis infection; TPT, tuberculosis preventive therapy; Mfx, moxifloxacin; Ofx, ofloxacin; Lfx, levofloxacin; E, ethambutol; Eth, ethionamide; H, isoniazid; Cfx, ciprofloxacin; Z, pyrazinamide; R, rifampicin; TB, tuberculosis; 95% CI, 95% confidence interval and NR, not reported.

**Table 2 pathogens-11-01189-t002:** Available guidelines on the preventive therapy for DR-TB contacts.

Source	Year of Publication	Population Addressed	Recommendation to Treat	Watchful Observation Approach	Drug	Ancillary Drugs	Treatment Duration
WHO	2020	General	Yes	Consider	Lfx	E, Eth	6 months
ECDC	2012	General	Yes	Consider	Lfx	No	6 months
ATS/CDC/ ERS/IDSA	2019	General	Yes	Not recommended	Lfx	No	6–12 months
MSF	2022	Pediatric	Yes	Consider	Lfx	No	6 months

Abbreviations: WHO, World Health Organisation; ECDC, European Centre for Disease Prevention and Control; ATS, American Thoracic Society; CDC, U.S. Centers for Disease Control and Prevention; ERS, European Respiratory Society; IDSA, Infectious Diseases Society of America; MSF, Doctors Without Borders; Lfx, levofloxacin, E, ethambutol and Eth, ethionamide.

**Table 3 pathogens-11-01189-t003:** Trial landscape on preventive therapy for DR-TB contacts.

Study	Status	Population Type	Structure	Duration of Treatment	Country	Total Population Size (N)	Duration of Follow-Up
V-QUIN	Enrolment completed	Adults > 15 years	Lfx vs. Placebo	6 months	Vietnam	3344	30 months
TB-CHAMP	Enrolment ongoing	Children < 5 years	Lfx vs. Placebo	6 months	South Africa	1556	24 weeks
PHOENIx	Enrolment ongoing	Adults > 15 years	Dlm vs. H	6 months	International	5610	26 weeks

Abbreviations: Lfx, levofloxacin; Dlm, delamanid and H, isoniazid.

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
