# Peer review of "Preventive Therapy for Contacts of Drug-Resistant Tuberculosis"

_pathogens, 2022, doi:10.3390/pathogens11101189_

Round 1

Reviewer 1 Report

It is a well written and interesting review on a very important topic. The English is fine and the text is very easy to read. I just have a small comment - take care the abbreviations - for example the the abbreviation LTBI is used without showing what they mean - line 72 pag 2 - it would be better to write what the things mean for the reader.

Author Response

Reviewer 1

It is a well written and interesting review on a very important topic. The English is fine and the text is very easy to read. I just have a small comment - take care the abbreviations - for example the the abbreviation LTBI is used without showing what they mean - line 72 pag 2 - it would be better to write what the things mean for the reader.

Answer: We thank the reviewer for his/her thoughtful review of our work and kind words. LTBI was defined in the Introduction section : “Once this happens, TB infection can be cleared by the innate immune system and remain silent – what we define as latent tuberculosis infection (LTBI) – or evolve to active TB disease [4]” (line 35-37).

Reviewer 2 Report

There’s ample evidence suggesting that preventive antibiotic regimens administered to people who are in close contact with diagnosed tuberculosis cases are effective and may facilitate TB control. These regimens, which in most cases comprise isoniazid and one rifamycin analog, are expected to provide little to no protection if the person was exposed to mono- or multi-drug resistant TB bacilli. Currently, there lacks a gold standard by which preventive therapies should be administered, and to my knowledge there hasn’t been a review which summarizes the recent development in the field since several of the seminal works (e.g., the clinical studies performed by Malik and others in Pakistan where MDR-TB is prevalent.) were only published within the past two or three years.  

In this draft, Kherabi and others discuss the current state of clinical research and implementations of preventive therapies for contacts of drug-resistant tuberculosis (DR-TB). The authors have re-organized and presented data from both real-world studies and mathematical modeling to address the potential and the urgency of implementing community preventive therapies for DR-TB, and concluded that fluoroquinolone-containing preventive regimens are, in general, safe to use and are likely to be effective. They elaborated on the insights from relevant clinical trials and the limitations thereof. Overall, this review has nicely presented the state of the art in clinical research of DR-TB preventive therapies, and has provided a comprehensive and up-to-date summary of a subject that’s utterly important yet somewhat under-studied. The draft is very well written, and the summary tables are well organized and clear to read. It is expected that this manuscript will be found interesting by a large group of researchers from the TB community, and I very much forward to reading it again in its published format. 

Several comments: 

Preventive therapy itself could create a niche that favors drug-resistant bacilli, which is an important factor to be considered for regimen design. These concerns have been extensively discussed by previous works, e.g. PMID 23576815. I would encourage the authors to discuss the trade-off between disease prevention and the promotion of DR-TB on persons at low to moderate risks.  

Some of the novel regimens with newly approved TB drugs have been tested for disease prevention in mouse models. For instance, this recent work by Kaushik and others (PMID: 34939891) has demonstrated the potential of short-term, bedaquiline-containing preventive therapy. These drugs are predicted to be active against DR-TB and should therefore be discussed in this review.   

There’s one extra space between “infection is” and “space” in line 155. 

Author Response

Reviewer 2

There’s ample evidence suggesting that preventive antibiotic regimens administered to people who are in close contact with diagnosed tuberculosis cases are effective and may facilitate TB control. These regimens, which in most cases comprise isoniazid and one rifamycin analog, are expected to provide little to no protection if the person was exposed to mono- or multi-drug resistant TB bacilli. Currently, there lacks a gold standard by which preventive therapies should be administered, and to my knowledge there hasn’t been a review which summarizes the recent development in the field since several of the seminal works (e.g., the clinical studies performed by Malik and others in Pakistan where MDR-TB is prevalent.) were only published within the past two or three years.  

In this draft, Kherabi and others discuss the current state of clinical research and implementations of preventive therapies for contacts of drug-resistant tuberculosis (DR-TB). The authors have re-organized and presented data from both real-world studies and mathematical modeling to address the potential and the urgency of implementing community preventive therapies for DR-TB, and concluded that fluoroquinolone-containing preventive regimens are, in general, safe to use and are likely to be effective. They elaborated on the insights from relevant clinical trials and the limitations thereof. Overall, this review has nicely presented the state of the art in clinical research of DR-TB preventive therapies, and has provided a comprehensive and up-to-date summary of a subject that’s utterly important yet somewhat under-studied. The draft is very well written, and the summary tables are well organized and clear to read. It is expected that this manuscript will be found interesting by a large group of researchers from the TB community, and I very much forward to reading it again in its published format. 

Several comments: 

Preventive therapy itself could create a niche that favors drug-resistant bacilli, which is an important factor to be considered for regimen design. These concerns have been extensively discussed by previous works, e.g. PMID 23576815. I would encourage the authors to discuss the trade-off between disease prevention and the promotion of DR-TB on persons at low to moderate risks.  

Answer: We thank the reviewer for his/her thoughtful review of our work and kind words. We agree with the reviewer and we added a discussion on the trade-off between disease prevention and the promotion of DR-TB on persons at low to moderate risks (lines 155-161).

Some of the novel regimens with newly approved TB drugs have been tested for disease prevention in mouse models. For instance, this recent work by Kaushik and others (PMID: 34939891) has demonstrated the potential of short-term, bedaquiline-containing preventive therapy. These drugs are predicted to be active against DR-TB and should therefore be discussed in this review.   

Answer: We thank the reviewer for his/her suggestion. We added a sentence on this specific example (lines 241-244) : “Moreover, recent studies on mouse models have demonstrated the promising potential of short-term regimens based on newly approved drugs such as bedaquiline, which should be evaluated as a possible candidate in DR-TB TPT [45].”

There’s one extra space between “infection is” and “space” in line 155.  

Answer: We thank the reviewer for his/her correction. We deleted this extra space.